# Examining the Theoretical Relationship between Constructs in the Person-Centred Practice Framework: A Structural Equation Model

**DOI:** 10.3390/ijerph182413138

**Published:** 2021-12-13

**Authors:** Tanya McCance, Brendan McCormack, Paul Slater, Donna McConnell

**Affiliations:** 1Istitute of Nursing and Health Research, Ulster University, Belfast BT37 0QB, UK; tv.mccance@ulster.ac.uk (T.M.); pf.slater@ulster.ac.uk (P.S.); 2Centre for Person-Centred Practice Research, Queen Margaret University, Edinburgh EH21 6UU, UK; bmccormack@qmu.ac.uk

**Keywords:** person-centred practice, structural equation model, survey

## Abstract

Research relating to person-centred practice continues to expand and currently there is a dearth of statistical evidence that tests the validity of an accepted model for person-centred practice. The Person-centred Practice Framework is a midrange theory that is used globally, across a range of diverse settings. The aim of this study was to statistically examine the relationships within the Person-centred Practice Framework. A cross sectional survey design using a standardized tool was used to assess a purposive sample (*n* = 1283, 31.8%) of multi-disciplinary health professionals across Ireland. Survey construct scores were included in a structural model to examine the theoretical model of person-centred practice. The results were drawn from a multi-disciplinary sample, and represented a broad range of clinical settings. The model explains 60.5% of the total variance. Factor loadings on the second order latent construct, along with fit statistics, confirm the acceptability of the measurement model. Statistically significant factor loadings were also acceptable. A positive, statistically significant relationship was observed between components of the Person-centred Practice Framework confirming it’s theoretical propositions. The study provides statistical evidence to support the Person-centred Practice Framework, with a multidisciplinary sample. The findings help confirm the effectiveness of the Person-Centred Practice Index for-Staff as an instrument that is theoretically aligned to an internationally recognised model for person-centred practice.

## 1. Introduction

The value placed on person-centredness as the preferred approach in health and social care is uncontested, as evidenced in policy and strategy development globally [1,2,3]. The knowledge base underpinning person-centredness continues to expand, with greater clarity on person-centredness as a concept relevant to international healthcare, and an increased understanding of the key components that need to be considered for effective implementation of person-centred practice [4,5,6]. This has led to the development of conceptual frameworks and models depicting the components of person-centredness (e.g., [7,8]), alongside the development of tools to enable measurement [9]. This paper presents the outcomes from a programme of work that has focused on the development of a conceptual framework for implementing person-centred practice, the Person-centred Practice Framework (PCPF) [3,10,11], and development of a tool aligned to the framework that can offer a valid and reliable standardised measure, the Person-centred Practice Inventory for Staff (PCPI-S) [12]. It is acknowledged in the literature that many instruments developed to measure person-centredness use proxy indicators and often do not provide a direct measurement of the concept of person-centred practice. The data generated from the PCPI-S has provided a unique opportunity to use data to further test theoretical relationships within the PCPF. This paper contributes to a limited evidence base that illustrates relationships within a mid-range theory through structural equational modelling.

### Background

Person-centredness reflects the ideals of humanistic caring in which there is a moral component, and the practice has at its basis a therapeutic intent, which is translated through relationships that are built on effective interpersonal processes. It is generally accepted that the principles underpinning person-centredness as an approach focuses on: treating people as individuals; respecting their rights as a person; building mutual trust and understanding; and developing positive relationships [11]. This philosophical position reflects models for developing person-centred healthcare that fundamentally take account of the humanness of people. The challenge, however, continues to be how these principles are translated into everyday practice to enable multiprofessional teams to deliver this standard of care consistently over time [13,14]. The promotion of ‘person-centredness’ is consistent with international developments in health-care policy. Whilst person-centredness has increasingly been positioned as a central focus in nursing and health care, theoretical developments have largely focused on related concepts and theories, especially relationship-centred care [15], compassionate care [16] and dignified care [17]. However, none of these particular theories adopt a truly holistic perspective that embraces person-centredness as a unifying theory of how all persons (healthcare workers and patients/families/communities) experience health care. Further, these perspectives have not stood the test of time as ‘alternatives’ but instead are increasingly seen as components of person-centred nursing and healthcare, or as constructs that explain different dimensions of person-centredness. This goes some way to affirming the importance of person-centred approaches, not just as care practices in particular professional groups, but as a philosophical underpinning of health-care systems that places people at the centre. The placing of people at the centre of care systems is exemplified by the World Health Organization, which has set out a comprehensive framework of people-centred health services [18]. The Person-centred Practice Framework [11], whilst embracing these theoretical perspectives also adopts a whole-systems approach to the practice of person-centred healthcare and operates at the levels of individual people, communities and populations.

The Person-centred Practice Framework (PCPF) was originally born out of a desire to operationalise person-centredness in a way that would illuminate the practice, and provide practitioners with a language that would enable them to name components of person-centredness and the barriers and enablers that influence its development in the workplace. The development of the Framework has spanned over a decade and whilst it has its origins in nursing practice and is described in a nursing version of the framework [10], the PCPF is situated within healthcare systems more broadly [3]. The latest version of the Framework [3] reflects changes made over time to ensure applicability to a wide range of healthcare workers across multiple contexts. It continues to be tested and refined through an ongoing programme of applied research within a multidisciplinary context.

The PCPF [3] comprises four main domains which are briefly described below. 

Prerequisites focus on the attributes of staff and include: being professionally competent, having developed interpersonal skills, being committed to the job, being able to demonstrate clarity of beliefs and values, and knowing self.The practice environment reflects the complexity of the context in which healthcare is experienced and includes: appropriate skill mix; systems that facilitate shared decision making; the sharing of power; effective staff relationships; organisational systems that are supportive; potential for innovation and risk taking; and the physical environment.Person-centred processes focus on ways of engaging that are necessary to create connections between persons, which include: working with the person’s beliefs and values; engaging authentically; being sympathetically present; sharing decision making; and working holistically.The outcome from the development of effective person-centred practice is a healthful culture that enables human flourishing.

The framework sits within a fifth domain, described as the *macro context*, which reflects the factors that are strategic and political in nature that influence the development of person-centred cultures and include: health and social care policy; strategic frameworks; workforce developments; and strategic leadership [11].

The relationship between the four main domains of the PCPF is represented pictorially, that being, to reach the centre of the framework, the attributes of staff must first be considered, as a prerequisite to managing the care environment, in order to provide effective care through the care processes. This ordering, ultimately leads to the achievement of the outcomes—the central component of the framework (Figure 1). It is also acknowledged that there are relationships between the constructs. The original PCPF was described as a mid-range theory [19]. It’s place on the continuum of theory development was made explicit by McCormack & McCance [3,10] drawing on the seminal work of Fawcett [20]. Fawcett argues that mid-range theories articulate one or more relatively concrete and specific concepts that are derived from a conceptual model. Furthermore, the propositions that describe these concepts propose specific relationships between them, and empirical indicators provide the means of measuring concepts within a mid-range theory. The PCPF meets the criteria for a mid-range theory, in that it has been derived from two abstract conceptual frameworks [21,22], comprises concepts that are relatively specific, and outlines relationships between the concepts. 

Furthermore, recent advancements have been made to develop empirical indicators to measure concepts within the PCPF. The development of the Person-centred Practice Inventory—Staff (PCPI-S) provides a measure that effectively maps directly onto the constructs of the PCPF. De Silva [9] conducted a review of existing tools to measure person-centred practice and reported that of the 114 instruments identified, none provided direct measurement of the concepts of person-centred practice. Many quantitative tools relied on statistical techniques that draw heavily on a posteriori classification of constructs that emerge from reductionist techniques such as exploratory factor analysis.

The PCPI-S holds to the concept of theory driven instrument development [12]. Instrument items map directly to accepted definitions that make up the model: factors onto items and second order latent variables onto first order factors. It provides quantitative data to allow testing of the theoretical model that underpins it. The comprehensiveness of the PCPF and as a consequence its complexity, gives the model an advantage over other simplistic models that reduce person-centred practice to measurements of broad, ill-defined concepts. These use, at best, proxy measures and add little to advance understanding of how to effectively measure and move toward person-centred outcomes. 

## 2. Materials and Methods

A quantitative cross sectional survey research design using the PCPI-S was used to generate sufficient data to adequately test the measurement model. The PCPI-S is a 59-itemed instrument that measures 17 constructs aligned to the Person-centred Practice Framework [11]. All items of the PCPI-S are measured on a five-point scoring range (1—Strongly Disagree to 5—Strongly Agree), with higher scores indicating higher levels of agreement. Demographic details of participants were also collected. The relationship between the items to first order constructs and consequently, first order to second order, is provided in the factor structure and psychometric properties of the PCPI-S reported by Slater et al [12] and Bing-Jonsson et al [23] and were acceptable.

### 2.1. Sample

The PCPI-S was tested with a sample of health professionals drawn from six organisations representing 14 acute hospital settings and 45 clinical units in Northern Ireland and Republic of Ireland. Organisations, hospital settings and clinical sites were all self-nominating to participate in the various studies, using the PCPI-S, to examine the current state of person-centred practice in Ireland. A range of clinical services were invited to participate in the overall study, including: Adult Services; Children and Young People; Primary Care and Older People; Mental Health, Emergency Department, Learning Disability, and Maternity. The settings were selected so as to obtain a good representation of views on person-centred practice for health professionals across the participating organisations. A gatekeeper in each clinical unit randomly distributed questionnaire packs to all health professionals working within their unit. The following inclusion criteria was applied: (i) working full time; (ii) have worked in the clinical setting for at least 6 months; and (iii) willing to participate. A response rate of 31.8% (*n* = 1283) was achieved (Table 1). Corresponding data set is available on request.

### 2.2. Ethical Approval

Full ethical approval was sought and gained from the Ulster University Institute of Nursing and Health Research committee and the relevant organisational and ethical bodies in line with research governance requirements. The main ethical issues related to informed consent and assuring confidentiality and anonymity for all participants. 

### 2.3. Procedure

Questionnaires were distributed to the gatekeeper in each clinical setting with the consent of the unit nurse manager/clinical nurse manager who provided a list of the total population of health professionals in the settings. A total of 4039 questionnaire packs were distributed across clinical settings. Participants were asked to complete the questionnaire and return it in the envelope provided for collection by a researcher (Data collected March–December 2015). The process of implied consent was made explicit in the Participant Information Sheet, whereby a completed questionnaire implied consent to participate. This ensured the confidentiality and anonymity of returned questionnaires. A deadline of two weeks was given for the return of questionnaires and a week of follow-up visits to collect completed questionnaires. Questionnaires collected were collated and categorised for data analysis by construct and clinical setting.

### 2.4. Measurement Model

Construction of the measurement model for testing was based on the PCPF (Figure 1). The 59 items of the PCPI-S map onto the 17 constructs of the framework, and the constructs onto second order latent factors [12,13]. The relationship between the items and their unobserved first order latent variables assumes the first order latent variables drives the indicators (i.e., these indicators are assumed to be observable instances or manifestations of their unobserved latent variables, and thus changes in the unobserved latent variables are ‘indicated’ by observable changes in the items in its measure [23]. The relationship between item and first order unobserved latent variable psychometric properties has been previously presented [12,13]. 

Similarly, a second order latent variable assumes the first order latent variables to be ‘driven’ by the second order latent variable and its subsequent filter down to the items. Wang and Wang [24] suggest that the aggregation of items to first order, and eventually to broad second order latent variables, allows for complex models to be simplified and interaction effects between the second order variables examined. This is the method used to examine and test the PCPF model. In this paper, structural modelling using actual scores for each construct was used to fit the second order latent variable measurement model and the interaction between the second order variables. The PCPF model postulates a direct relationship between second order latent variables Prerequisites and the Care Environment, and subsequently Care Processes. This measurement model (Figure 2) was tested as the initial model. Given the relative newness of the PCPI-S and the lack of comparative findings relating to the testing and fitting of the PCPF, a pragmatic approach to the specification of the model was applied.

### 2.5. Statistical Analysis

The data were prepared in line with previous instruction: Descriptive and measures of dispersion statistics were generated for all items to help inform subsequent analysis. The 59-items were summated to 17 constructs as per guidelines [12,13]. Inter-item correlations were generated to examine for collinearity prior to full analysis. Confirmatory factor analysis was used to examine the theoretical measurement model. Examination of the data indicates kurtosis on a few of the items. Therefore, as a precaution, the data were analysed using maximum likelihood robust (MLR) as relevant with continuous and skewed data. 

The model was re-specified using the modification indices provided in the statistical output until acceptable and a statistically significant relationship identified. All re-specifications of the model were guided by principles of meaningfulness (a clear theoretical rationale); transitivity (if A is correlated to B, and B correlated to C, then A should correlate with C); generality rule (if there is a reason for correlating the errors between one pair of errors, then all pairs for which that reason applies should also be correlated) [25].

Acceptance modification criteria included the following: within factors inter-item correlated errors were permitted and based on criteria of being theoretically relevant introduced one at a time and selected on highest score first (exceeding scores of 3.98).across factors inter-item correlated errors were permitted and based on criteria of being theoretically relevant introduced one at a time and selected on highest score first (exceeding scores of 3.98)only statistically significant relationship retained to help produce as parsimonious a model as possible.

Acceptable fit statistics were set at root mean square estimations of approximation (RMSEA) of 0.08 or below; 90% RMSEA higher bracket below 0.08; and confirmation fit indices (CFI) of 0.90 or higher, Tucker–Lewis Index (TLI) above 0.90 and standardised root mean squared residual (SRMR) below 0.08 [24,26]. Hair et al. [27] advocate a factor loading of 0.30 for a sample size of greater than *n* = 350.

## 3. Results

A breakdown of the demographic details is outlined in Table 1. There was a good spread of responses across banding and experience. There was an uneven distribution across healthcare settings in the total sample as the organisations requested different samples to be surveyed. Whilst the majority of respondents were from a nursing background, there was representation across all health professions.

All 17 constructs were positively scored at ‘Agreed’ (Table 2). The lowest scored construct was ‘Shared Decision Making Systems’ and the highest was ‘Providing Holistic Care’. The measures of skewness and kurtosis were acceptable and indicate a normality of distribution with the exception of a few points of deviation on kurtosis (see Table 2).

The correlation matrix scores (see Table 3) indicate a positive correlation between the 17 measures at a low to moderate size. There were no issues of collinearity between construct scores.

### 3.1. Theoretical Framework

The Kaiser–Meyer–Olkin Measures of sampling adequacy 0.932; Bartletts test of sphericity (chi square = 11.41, df = 136, *p* = 0.001). KMO scores of ≥0.8 and Bartletts test of significant *p* ≥ 0.05 are acceptable values and indicate the appropriateness of the items prior to factor analysis. The original model failed to have acceptable fit statistics (see Table 4) and modifications to the original model were required. 

### 3.2. Correlated Error of Variances

A series of correlated errors were introduced to the model in an iterative process using the criteria outlined. This process continued to a position where the fit statistics were acceptable. The presence of correlated errors within factors indicates the principle of transitivity; generality and meaningfulness. There were five correlated error variance corrections included in the model. The five correlated errors of variance indicate a presence of an unmeasured source of influence (generality) and/or may be accounted from with justification of the relative closeness in the factors in the model (meaningfulness). The issue of generality does indicate an incomplete measurement model and that the questionnaire would be improved with the inclusion of factors that may not be adequately addressed in this model. The process of instrument modification resulted in acceptable fit statistics. 

A three-factor model explains 60.47% of the variance. All constructs were statistically significant. Examination of the Cronbach’s alpha scores indicate acceptable factor structures: Prerequisites Cronbach’s alpha = 0.77; Care Environment = 0.75; and Care Processes = 0.88 and are acceptable. Examination of the factor loadings show the constructs ‘Developed Interpersonal Skills’ and ‘Being committed to the job’ offer the largest contribution to ‘Prerequisite’ and ‘Clarity of beliefs and values’ as least significant (See Table 5). ‘Shared Decision-making’ was the most significant predictor of ‘Care environment’ and all seven constructs contributed significantly to Care environment. All five constructs contributed significantly to ‘Care Processes’ but ‘Sympathetic Presence’ was the main contributor.

### 3.3. Relationship between Second Order Latent Variables

Examination of the relationship between the second order latent factors show Prerequisites on Care Environment 0.73 (SE = 0.027, EST/SE = 27.45, *p* = 0.00); Prerequisites on Care Processes 0.83 (SE = 0.043, EST/SE = 19.36, *p* = 0.00). Care Environment on Care Processes 0.08 (SE = 0.05, EST/SE = 1.79, *p* = 0.07). A standard unit change in Prerequisites will produce a 0.73 unit change in Care Environment; a 1 unit change in Prerequisites will produce a 0.83 unit change in Care Processes (See Figure 2). There was no significant relationship between care environment and care process.

‘Shared Decision-making’ was the most significant predictor of ‘Care environment’ and all seven constructs contributed significantly to Care environment. All five constructs contributed significantly to ‘Care Processes’ but ‘Sympathetic Presence’ was the main contributor.

## 4. Discussion

Person-centred practice provides a central tenet underpinning health and social care internationally [1,2]. The Person-centred Practice Framework [11] is internationally recognised and is being implemented globally [28]. This paper provides, for the first time, statistical evidence of the relationships between the theoretical constructs of the framework and what these might mean for developing our understanding of person-centredness and its operationalisation in healthcare practice. Whilst the data reinforce the importance of the constructs of the framework and their relevance to person-centred healthcare, a number of issues are raised by the findings of the study.

The significance of interpersonal skills as a core component of person-centred practice is reinforced by the findings. The constructs, *Developed Interpersonal Skills* and *Being committed to the job*, offer the largest contribution to the *Prerequisites* domain. The prerequisites of the Person-centred Practice Framework focus on the attributes of staff and are considered the key building blocks in the development of healthcare workers who can deliver effective person-centred care [11]. The fact that interpersonal skills and commitment to the job emerge as the most significant qualities is an illuminating finding. In their writing, McCormack and McCance [10] argue that person-centredness is essentially a relational practice that is dependent on well-established interpersonal skills that can be operationalised in different practice contexts. The significance of interpersonal relationships also reinforces the idea that person-centredness is more than the ‘doing’ of particular practices, but is much more about a way of ‘being’ as a practitioner. The dominant focus in healthcare developments on ‘doing’ person-centredness is one of the reasons why we continue to see problems in practice. Despite much investment in quality improvement work targeting the development of person-centred practice we still see, at best, what Laird et al [29] have described as ‘person-centred moments’. *Commitment to the job* further reinforces the being of person-centredness defined as: *‘*the dedication of practitioners demonstrated to patients, families, and communities through intentional engagement that focuses on achieving the best possible outcomes’ [3] (p. 55). Dedication implies presence and being in the moment with patients, families and communities and through such presence enabling engagement. The findings of this study reinforce the need for practitioners to have well-developed interpersonal skills that will enable them to be present with patients and families. 

The fact that *clarity of values and beliefs* had the least significance in explaining the theory, raises important issues about how person-centred practice cultures are developed. The data show that values and beliefs are important and of course are essential to the being of the person-centred practitioner. However, the data also suggest that having clear beliefs and values is not enough in itself for the delivery of person-centred healthcare. In contemporary healthcare there is significant emphasis placed on values and beliefs among healthcare teams through programmes such as ‘Values in Action’ [see https://www.hse.ie/eng/about/our-health-service/values-in-action/valuesinactionblog/how-the-nine-values-in-action-behaviours-were-developed.html for further details (accessed on 21 November 2021) that reinforce the importance of particular values being evident in healthcare practice behaviours. What such programmes fail to recognise, however, are the complex factors that have to be addressed on a continuous basis for such values to be translated into everyday practice. Emancipatory and transformational practice development methodologies [5], however, have played a significant role in demonstrating the need for continuous facilitated meaningful engagement to develop healthful cultures. Evidence from practice development programmes illustrate how clarity of beliefs and values acts as a foundation for culture change, leadership development, team effectiveness and consistency in patient care [30,31]. Notably, such programmes also depend on long-term engagement for the development and embedding of culture change in healthcare settings—it is not a quick-fix solution. This is both a strength and a weakness of practice development as in a fast-moving healthcare context, quick-fix solutions that are the artefacts of person-centredness will always be favoured. Furthermore, the findings from this study suggest that this focus needs to be challenged if we are to see a large-scale shift towards person-centred cultures in healthcare organisations.

The data also suggests that the *shared decision-making systems* construct of the *practice environment* domain was the most likely predictor of a person-centred culture. This finding reinforces the importance of interdisciplinarity and service-user participation in healthcare practice. An organisational commitment to collaborative, inclusive and participative ways of engaging within and between teams is essential for person-centred practice [3]. Shared decision-making among team members is the foundation of interdisciplinary practice [32], and the essence of person-centred healthcare. Ensuring that service-users play a key role in shaping their care is fundamental to person-centred practice and contributes to person-centred outcomes [33,34]. Research by Ekman and colleagues [33,34] shows that when patients are active agents in the development of a care plan, when healthcare teams collaborate to ensure the implementation of the plan, and when evaluation of the impact of the plan is undertaken from the perspective of the patient, then patient and team outcomes can be demonstrated. The findings from the research reported in this paper reinforce the importance of this collaborative working and illustrate engagement and negotiation in action. 

From the perspective of the *person-centred processes,* it is illuminating that the data suggest that *being sympathetically present* is core to all of the other person-centred processes. McCance & McCormack [3] (p.57) define *being sympathetically present* as: “an engagement that recognises the uniqueness and value of the patient by appropriately responding to cues that maximise coping resources through the recognition of important agendas in the person’s life”. The statistical significance of sympathetic presence in this research further highlights the need for the professional development of healthcare practitioners to focus on developing their ‘ways of being’, alongside the focus on developing their professional competence. *Being sympathetically present* is a phenomenological process that reinforces the individuality of persons, which is core to person-centred practice. The evidence from the research reported in this paper reinforces that *being sympathetically present* is an essence of being person-centred and underpins all practices. This is a significant finding from this research as it begins to surface issues that healthcare organisations may need to address for person-centred cultures to become the norm. This includes: the reorganisation of services; challenging the standardisation agenda and protocolised care; and providing support systems for practitioners to enable them to sustain such individualised and engaged ways of being.

The findings from the study provide new data relating to the operalisation of person-centredness in healthcare. It also provides insights as to how issues relating to the translation of person-centred principles into practice can be addressed as identified by Moore et al. [13] and Sharp and colleagues [14]. The evidence suggests that all the theoretical constructs make a statistically significant contribution to the overall understanding of person-centred practice, but with varying degrees of significance. Importantly, the findings highlight the focus on specific aspects of the PCPF, identifying areas for change that can facilitate the development of person-centred cultures through a continuous focus on quality improvement programmes.

The findings add to a growing evidence base for a psychometrically sound tool [12,22], that maps onto an established theoretical framework, redressing De Silva’s [9] critique of tools to measure person-centred practice. The PCPI-S has shown its value in measuring person-centred practice [35]. The findings from this study continue to demonstrate that the PCPI-S has acceptable psychometric properties and displays the usefulness of the instrument as a means of informing how a theoretically driven approach to developing person-centredness in healthcare can be systematically developed through targeted interventions. The findings evidence the role that *prerequisites* (staff attributes) play in shaping the *practice environment* and engaging in *person-centred processes*. This is as expected in McCance and McCormack’s [11] theoretical framework and indicates that a focus on *prerequisites* can produce significant changes in shaping the overall approach to realising healthful cultures. 

### Limitations

Only minor modifications (correlated errors) were required to the model to provide acceptable fit statistics. The presence of correlated errors indicates the presence of influential and as yet unmeasured elements. However, the Person-centred Practice Framework [11] contains elements (e.g., the macro context) that are not measured by the PCPI-S and should be included in modifications to the instrument. This would provide a comprehensive assessment of person-centred practice and further help with the translational process. The theoretical framework requires further testing with varying samples, to further explore its potential and psychometric properties.

## 5. Conclusions

Person-centred practice is an internationally recognised indicator of good practice and the PCPF is a strong theoretical model for realising this in everyday practice. This paper provides statistical evidence to support the PCPF and uses an instrument that appears to effectively measure the relationships between the constructs underpinning the theory. This is one of the first studies that has attempted to articulate the relationship between person-centred constructs informed by an established theory. Ongoing research is needed to further test these relationships and build a reputable model of quality improvement that is theoretically driven and informs the development of targeted interventions that fulfil the need to be context-specific, but replicable across organisations and international contexts.

## Figures and Tables

**Figure 1 ijerph-18-13138-f001:**
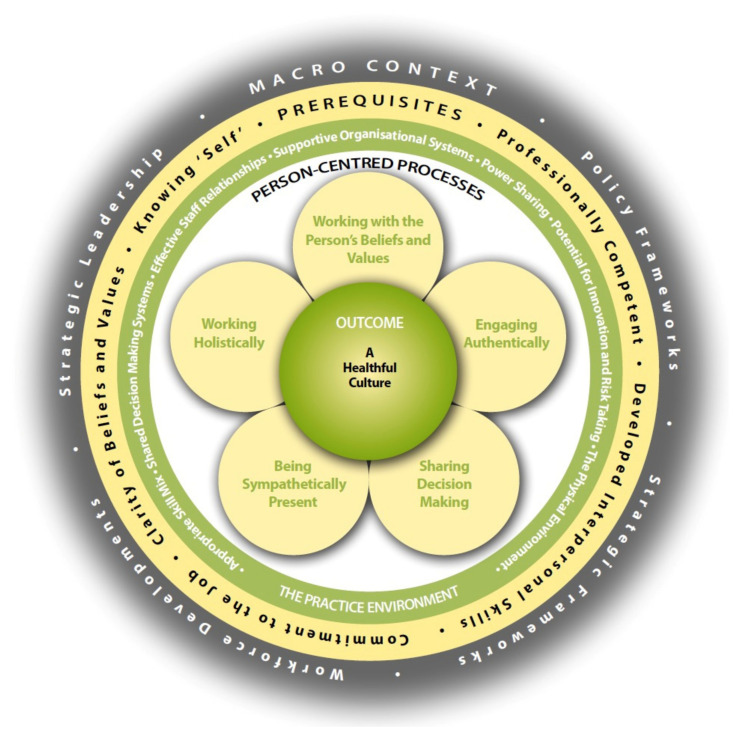
The Person-centred Practice Framework [11].

**Figure 2 ijerph-18-13138-f002:**
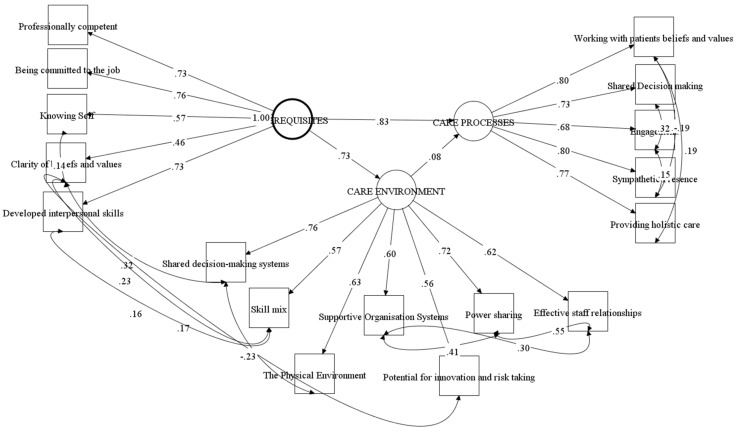
The Person-centred Practice Framework as a theoretical model.

**Table 1 ijerph-18-13138-t001:** Demographic spread of overall sample.

Profession		Experience		Nurses Only	
Nursing	84.4% (*n* = 1088)	<1 year	3.6% (*n* = 46)	Band 5	64.1% (431)
Medical	6.7% (*n* = 86)	1–5 years	24.1% (*n* = 305)	Band 6	20.1% (135)
Allied Health Professional	7.0% (*n* = 90)	6–10 years	15.7% (*n* = 199)	Band 7	14.6% (98)
Health Care Assistant	1.9% (*n* = 25)	Over 10 years	56.5% (*n* = 715)	Band 8	1.2% (8)

**Table 2 ijerph-18-13138-t002:** Mean scores and measures of distribution for constructs.

Code	CONSTRUCTS	Mean	SD	Skewness	Kurtosis	Alpha
	PREREQUISITES	4.16	0.37	−0.46	1.89	0.77
V1	Professionally Competent	4.24	0.46	−0.50	1.55	0.48
V2	Developed Interpersonal Skills	4.32	0.43	−0.34	0.76	0.67
V3	Being committed to the job	4.39	0.47	−0.75	0.75	0.74
V4	Knowing Self	3.96	0.58	−0.77	1.32	0.63
V5	Clarity of beliefs and values	3.90	0.58	−0.85	2.69	0.62
	THE CARE ENVIRONMENT	3.76	0.51	−0.70	1.20	0.75
V6	Skill mix	4.15	0.51	−0.65	1.51	0.40
V7	Shared decision making systems	3.55	0.78	−0.61	0.264	0.75
V8	Effective staff relationships	3.94	0.76	−1.15	1.80	0.83
V9	Power sharing	3.78	0.74	−0.89	1.09	0.79
V10	Potential for innovation and risk taking	3.80	0.67	−0.57	0.95	0.87
V11	The Physical Environment	3.96	0.60	−0.72	1.50	0.84
V12	Supportive Organisation Systems	3.18	0.83	−0.46	−0.13	0.86
	CARE PROCESSES	4.18	0.44	−0.33	1.41	0.88
V13	Working with patients beliefs and values	4.06	0.56	−0.46	1.48	0.78
V14	Shared Decision making	4.09	0.58	−0.43	0.88	0.74
V15	Engagement.	4.20	0.47	−0.21	2.51	0.78
V16	Sympathetic Presence	4.21	0.51	−0.53	1.63	0.70
V17	Providing holistic care	4.30	0.53	−0.57	0.62	0.78

**Table 3 ijerph-18-13138-t003:** Correlation matrix of Constructs of PCPI-S (Pearson Product scores—all significant at *p* > 0.05).

	V1	V2	V3	V4	V5	V6	V7	V8	V9	V10	V11	V12	V13	V14	V15	V16	V17
V1	1																
V2	0.543	1															
V3	0.557	0.570	1														
V4	0.377	0.441	0.438	1													
V5	0.321	0.300	0.345	0.380	1												
V6	0.357	0.410	0.355	0.315	427	1											
V7	0.378	0.329	0.335	0.304	0.446	0.454	1										
V8	0.287	0.323	0.264	0.225	0.314	0.343	0.570	1									
V9	0.365	0.351	0.299	0.278	0.305	0.378	0.626	0.744	1								
V10	0.365	0.322	0.298	0.286	0.344	0.339	0.401	0.354	0.457	1							
V11	0.385	0.380	0.457	0.355	0.359	0.355	0.386	0.330	0.397	0.341	1						
V12	0.250	0.246	0.249	0.309	0.268	0.285	0.516	0.566	0.666	0.347	0.387	1					
V13	0.517	0.463	0.520	0.422	0.344	0.351	0.340	0.332	0.418	0.382	0.469	0.368	1				
V14	0.438	0.447	0.465	0.339	0.307	0.347	0.337	0.340	0.391	0.293	0.387	0.337	0.601	1			
V15	0.448	0.505	0.466	0.337	0.292	0.373	0.260	0.229	0.296	0.315	0.347	0.212	0.552	0.655	1		
V16	0.570	0.566	0.565	0.399	0.330	0.352	0.320	0.300	0.352	0.342	0.383	0.282	0.571	0.590	0.618	1	
V17	0.456	0.480	0.555	0.380	0.259	0.321	0.310	0.336	0.396	0.292	0.477	0.313	0.696	0.560	0.516	0.612	1

**Table 4 ijerph-18-13138-t004:** Fit statistics for alternative measurement models of the PCPI-S.

Model	RMSEA	90% RMSEA	CFI	TLI	SRMR
Original Model	0.079	0.075–0.083	0.889	0.870	0.076
Accepted Model	0.034	0.033–0.035	0.901	0.893	0.049

**Table 5 ijerph-18-13138-t005:** Factor loading of first order factors to second order factors.

CONSTRUCTS	Estimate	S.E.	Est./S.E.	Variance
PREREQUISITES				
Professionally Competent	0.73	0.02	36.73	47%
Developed Interpersonal Skills	0.73	0.02	39.19	47%
Being committed to the job	0.76	0.02	38.29	43%
Knowing Self	0.57	0.02	24.69	67%
Clarity of beliefs and values	0.46	0.03	14.44	79%
THE CARE ENVIRONMENT				
Skill mix	0.57	0.03	21.63	68%
Shared decision-making systems	0.76	0.02	39.64	42%
Effective staff relationships	0.62	0.03	24.96	61%
Power sharing	0.73	0.02	35.44	47%
Potential for innovation and risk taking	0.56	0.03	19.35	69%
The Physical Environment	0.63	0.03	24.60	60%
Supportive Organisation Systems	0.60	0.02	25.27	64%
CARE PROCESSES				
Working with patients beliefs and values	0.80	0.02	47.32	37%
Shared Decision making	0.73	0.02	39.07	47%
Engagement.	0.68	0.02	30.42	54%
Sympathetic Presence	0.80	0.02	50.18	36%
Providing holistic care	0.77	0.02	49.42	41%

## Data Availability

The data presented in this study are available on request from the corresponding author. The data are not publicly available due to ethical protocol.

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
