# Peer review of "Examining the Theoretical Relationship between Constructs in the Person-Centred Practice Framework: A Structural Equation Model"

_ijerph, 2021, doi:10.3390/ijerph182413138_

Round 1
Reviewer 1 Report
General comment:
- The authors have aimed to demonstrate that the Person-centred practice is an indicator of good practice and a strong theoretical model for healthcare and social practice. In terms of novelty, this study is an reinforcement of previous findings on this model. What is interesting is, this study contains a big sample size. However, the manuscript can still be further improved with the specific comments below.
Major comments:
- The introduction needs improvement in terms of establishing the novelty of your study. We know and agree that person-centredness is the preferred approach in health and social care. But what is the difference between your study and previous study other than the different community samples i.e. your study is from Northern Ireland and Republic of Ireland? I think this has to be clearly written and established in introduction again. If not, the whole study is like a repeat of past studies.
- It is important to establish your aim of study clearly. I failed to understand what your study is trying to achieve after reading your introduction. A paragraph summarising your literature reviews and then how these reviews lead to your hypothesis will be helpful.
- How do you categorise if a PCPI-S item is categorised into first or second order latent factors? Please explain clearly in the manuscript in results and materials and methods. Are there any references to support and validate your categorisation of these covariates instead of others? This is critical. Please justify these categorisations in a scientific manner both in results and in materials/methods.
- Figure 2 (The Person-centred Practice Framework as a theoretical model.) is pretty messy and some words are not visible at all. I cannot see the links of your ideas via the arrows. Some arrows are pointing in the air with no focal point. Please improve this diagram.
Minor comments:
- There are several careless grammatical and spelling slips which are not acceptable at this academic level. Please seek to edit and proofread your whole manuscript again for consideration for publication again.
Author Response
Reviewer 1
General comment:
- The authors have aimed to demonstrate that the Person-centred practice is an indicator of good practice and a strong theoretical model for healthcare and social practice. In terms of novelty, this study is an reinforcement of previous findings on this model. What is interesting is, this study contains a big sample size. However, the manuscript can still be further improved with the specific comments below.
Major comments:
- The introduction needs improvement in terms of establishing the novelty of your study. We know and agree that person-centredness is the preferred approach in health and social care. But what is the difference between your study and previous study other than the different community samples i.e. your study is from Northern Ireland and Republic of Ireland? I think this has to be clearly written and established in introduction again. If not, the whole study is like a repeat of past studies.
The introduction has been enhanced to address this comment.
The PCPI-S holds to the concept of theory driven instrument development [13].
- It is important to establish your aim of study clearly. I failed to understand what your study is trying to achieve after reading your introduction. A paragraph summarising your literature reviews and then how these reviews lead to your hypothesis will be helpful.
I have inserted an aim for the study in the abstract and at the end of the introduction.
- How do you categorise if a PCPI-S item is categorised into first or second order latent factors? Please explain clearly in the manuscript in results and materials and methods. Are there any references to support and validate your categorisation of these covariates instead of others? This is critical. Please justify these categorisations in a scientific manner both in results and in materials/methods.
I have clarified the items to constructs relationship in the methods section. Appropriate references to support and guide the readers relating to this relationship are provided). Line 107-108. This is further referred to in the measurement model (page 140 – 146).
- Figure 2 (The Person-centred Practice Framework as a theoretical model.) is pretty messy and some words are not visible at all. I cannot see the links of your ideas via the arrows. Some arrows are pointing in the air with no focal point. Please improve this diagram.
I have redressed this as much as possible. I am at the behest of the mplus software package when it comes to the diagram. I agree they can produce figures that are messy.
Minor comments:
- There are several careless grammatical and spelling slips which are not acceptable at this academic level. Please seek to edit and proofread your whole manuscript again for consideration for publication again.
I have reviewed the whole manuscript and made corrections as required.
Reviewer 2 Report
Thank you for allowing me to review your paper. The topic is an exciting and timely need. The overall flow of this manuscript suits well to the focus of the journal. However, some specific aspects and clarification of this manuscript need to be considered before the decision for publication. See below,
- In line 159, please check the punctuation for ie and closing parenthesis.
- You mentioned "interaction effects" between the second-order variables examined. A more specific explanation is needed to help better understand your paper.
- Why do you use only three fit indices for evaluating the hypothesized model? What about TLI and SRMR for this model?
- There is some confusion about the analysis and comments. In lines 180-181, you mentioned that you used MLR for the investigation because of the non-normality for many items. However, in line 209, when you say the assumption, you assumed the normality for the data. Would you clarify that?
- In Table 3, you need to be consistent in reporting the result. One result has a period, but the others do not. If you decide not to use decimal periods, please add the note to let the readers know the value.
- In your analysis, you examined the theoretical framework based on the existing theory with the data, right? Then, why did you use KMO and Barlett's test? Your explanation seems to be close to the confirmatory factor analysis rather than exploratory factor analysis. If you can add the reference to support your analysis, it will be ok.
- In your result, you mentioned prerequisites on care processes is .875. Based on this interpretation, the directional arrow is assumed to present the relationship, not the non-directional arrow in figure 2.
- In figure2, you did not present the nonsignificant relationship like care environment on care process. You can add all the results from your analysis in the figure with having the * for the significant relationship. Or you need to add "Note" to mention that only significant results are presented in the figure.
- In figure 2, based on your explanation, all variables are latent constructs. Why did you use rectangles, not oval, for the first-order constructs? And it would be better to use the construct names rather than v#. When you interpret the results, you use the construct names. There may lead to any confusion for the readers. If it is impossible to add all constructs' names in the figure, have a separate supplement to show all the v# indicates what.
Author Response
Reviewer 2
Thank you for allowing me to review your paper. The topic is an exciting and timely need. The overall flow of this manuscript suits well to the focus of the journal. However, some specific aspects and clarification of this manuscript need to be considered before the decision for publication. See below,
- In line 159, please check the punctuation for ie and closing parenthesis.
This has been clarified.
- You mentioned "interaction effects" between the second-order variables examined. A more specific explanation is needed to help better understand your paper.
This has been addressed in section line 151 and links in with the details provided subsequently.
- Why do you use only three fit indices for evaluating the hypothesized model? What about TLI and SRMR for this model?
Hooper et al recommend that three indices as sufficient to examine the model – as long as it included the RMSEA value. However for inclusiveness, additional information provided a typo corrected Line 179 – 181.
- There is some confusion about the analysis and comments. In lines 180-181, you mentioned that you used MLR for the investigation because of the non-normality for many items. However, in line 209, when you say the assumption, you assumed the normality for the data. Would you clarify that?
This has been clarified in lines 164 and 194 to indicate that some items deviated from normality of distribution and as a precautionary measure MLR analysis was sued.
- In Table 3, you need to be consistent in reporting the result. One result has a period, but the others do not. If you decide not to use decimal periods, please add the note to let the readers know the value.
This has been clarified in table 3.
- In your analysis, you examined the theoretical framework based on the existing theory with the data, right? Then, why did you use KMO and Barlett's test? Your explanation seems to be close to the confirmatory factor analysis rather than exploratory factor analysis. If you can add the reference to support your analysis, it will be ok.
Reference for inclusion of test included in line 210.
- In your result, you mentioned prerequisites on care processes is .875. Based on this interpretation, the directional arrow is assumed to present the relationship, not the non-directional arrow in figure 2.
This has been corrected in the new figure.
- In figure2, you did not present the nonsignificant relationship like care environment on care process. You can add all the results from your analysis in the figure with having the * for the significant relationship. Or you need to add "Note" to mention that only significant results are presented in the figure.
This has been corrected in the figure.
- In figure 2, based on your explanation, all variables are latent constructs. Why did you use rectangles, not oval, for the first-order constructs? And it would be better to use the construct names rather than v#. When you interpret the results, you use the construct names. There may lead to any confusion for the readers. If it is impossible to add all constructs' names in the figure, have a separate supplement to show all the v# indicates what.
This has been corrected in the figure.
Reviewer 3 Report
Dear authors, I appreciate the idea of the study and what its potential represents. The manuscript is fairly well written. However, I have some comments that I hope you will find useful to strengthen the paper.
Abstract
- construct scores by summing up the items scores - it is not essential information to be provided in the abstract
- the ethical approvement, as well
- abbreviation - to be avoided as there is no explanation for it
Keyword - cross-sectional survey - I don't see it as a keyword given the fact that the focus is not on that type of design
Background
- The manuscript, in its current form, needs improvement in the theoretical setting. Which theories support the framework? What other models/theories are described in the previous literature? How this framework differs from alternative models? What is the place of PCPF within other models?
- lines 56-57 - use of abbreviation, citation doesn't follow the guidelines of the journal
- lines 79-80 - be more specific about the four main components. Besides what we can see in Figure 1, which four components? Why there are four out of five more important?
Instruments
- lines 114 - 121: usually, there are presented examples of items from the instruments
Sample
- we find the sample size in the Procedure section. It should be mentioned in Sample section
- Besides what we can find in table 1 concerning the demographic data, a short description of the demographic dimensions should be presented.
Measurement model
- the labels of the constructs should be mentioned also in this section.
Results
- table 2 - the label for the last column is missing. It is alpha Cronbach values.
- table 3 - correlations? What kind of test/indices have you used? What indices do you report? Also, further details about V1, V2 etc. are needed.
- lines 243-246 - the indices should be added
- lines 251-253 - capital letter for P should be replaced by p
Discussion
- lines 344-350 - I would delete these lines. The information is obvious and common sense.
- line 359 - "Innovative" is a too strong word in that sentence.
Limitations
- this section should be enriched
References
- errors in following the MDPI reference formatting guide
Author Response
Reviewer 3
Dear authors, I appreciate the idea of the study and what its potential represents. The manuscript is fairly well written. However, I have some comments that I hope you will find useful to strengthen the paper.
Abstract
- construct scores by summing up the items scores - it is not essential information to be provided in the abstract – this has been addressed.
- the ethical approvement, as well. This has been removed.
- abbreviation - to be avoided as there is no explanation for it - rectified.
Keyword - cross-sectional survey - I don't see it as a keyword given the fact that the focus is not on that type of design – removed.
Background
- The manuscript, in its current form, needs improvement in the theoretical setting. Which theories support the framework? What other models/theories are described in the previous literature? How this framework differs from alternative models? What is the place of PCPF within other models?
- lines 56-57 - use of abbreviation, citation doesn't follow the guidelines of the journal addressed in situ
- lines 79-80 - be more specific about the four main components. Besides what we can see in Figure 1, which four components? Why there are four out of five more important?
I have clarified this issue.
Instruments
- lines 114 - 121: usually, there are presented examples of items from the instruments
the references to other published papers that explore these items specifically are provided throughout the paper
Sample
- we find the sample size in the Procedure section. It should be mentioned in Sample section
This has been moved as suggested.
- Besides what we can find in table 1 concerning the demographic data, a short description of the demographic dimensions should be presented.
Rectified in line 196.
Measurement model
- the labels of the constructs should be mentioned also in this section.
I have provided the label for all the constructs in Figure 2 and they are included in table 2 and table 4.
Results
- table 2 - the label for the last column is missing. It is alpha Cronbach values.
Corrected.
- table 3 - correlations? What kind of test/indices have you used? What indices do you report? Also, further details about V1, V2 etc. are needed.
Ive tied the V1, v2 etc into table 2 to aid clarification as inclusion of names would make table 3 too cluttered.
- lines 243-246 - the indices should be added
- lines 251-253 - capital letter for P should be replaced by p
All issues addressed in situ.
Discussion
- lines 344-350 - I would delete these lines. The information is obvious and common sense – These lines don’t tally in the edited version so not sure what is being requested to be deleted?? .
- line 359 - "Innovative" is a too strong word in that sentence – as above?
Limitations
- this section should be enriched
This has been strengthened.
References
- errors in following the MDPI reference formatting guide
references addressed.
Reviewer 4 Report
Some comments are suggested:
- The abstract must be structured with the sections of the manuscript.
- The objective in the abstract should be clarified and properly formulated.
- It would be interesting if the keywords are DeCS / MeSH descriptors.
- Although the introduction is adequate and interesting, the reason for the instrument used among the 114 existing ones should be justified, highlighting its usefulness or advantages. In addition, the last paragraph of the introduction should include the objective of the study clearly and appropriately formulated.
- The sample section does not indicate data on the universal or target population and what type of minimum sample calculation was required to achieve representativeness. In addition, it refers to the randomization of the units but not to what type of sampling was used for the selection of the sample. Some of this information is in the procedure section but it should be placed consistently or try to integrate both sections.
- It would be interesting to provide the validation data from previous studies on the instrument used.
- In table 1 AHP and HCA should be clarified and what does band mean
- In Table 2, the last column on the right does not indicate what it refers to, although it may be the p-value. Also, could it be justified why or what utility is included in including Skewness and Kurtosis?
- Table 4 must include a legend clarifying the abbreviations
- Figure 2 is not self-explanatory.
- In line 301 a link has been included that, although it may be interesting, must be put in parentheses explaining why it is included.
- The discussion should be reviewed regarding the use of full stops. It is advisable to avoid such long and extensive paragraphs, mixing some topics with others about the phenomenon.
Author Response
Reviewer 4
Some comments are suggested:
- The abstract must be structured with the sections of the manuscript.
Done.
- The objective in the abstract should be clarified and properly formulated.
This has been addressed
- It would be interesting if the keywords are DeCS / MeSH descriptors.
These are accurate descriptors of the study.
- Although the introduction is adequate and interesting, the reason for the instrument used among the 114 existing ones should be justified, highlighting its usefulness or advantages. In addition, the last paragraph of the introduction should include the objective of the study clearly and appropriately formulated.
The introduction has been enhanced to address this comment and aim of the study added.
- The sample section does not indicate data on the universal or target population and what type of minimum sample calculation was required to achieve representativeness. In addition, it refers to the randomization of the units but not to what type of sampling was used for the selection of the sample. Some of this information is in the procedure section but it should be placed consistently or try to integrate both sections.
- It would be interesting to provide the validation data from previous studies on the instrument used.
Reference to previous papers examining the psychometric properties are provided for readers to access. There is a lack of data to this specific to this type of work.
- In table 1 AHP and HCA should be clarified and what does band mean – Addressed in table 1.
- In Table 2, the last column on the right does not indicate what it refers to, although it may be the p-value. Also, could it be justified why or what utility is included in including Skewness and Kurtosis?
These are Cronbach alpha scores and Ive made that clear in the text.
- Table 4 must include a legend clarifying the abbreviations
These abbreviations are explained in the data analysis section.
- Figure 2 is not self-explanatory.
Have tried to clarify it more clearly now.
- In line 301 a link has been included that, although it may be interesting, must be put in parentheses explaining why it is included.
This has been done.
- The discussion should be reviewed regarding the use of full stops. It is advisable to avoid such long and extensive paragraphs, mixing some topics with others about the phenomenon.
I have reviewed the whole manuscript and made corrections as required.
Round 2
Reviewer 1 Report
Comments are addressed satisfactorily.
Reviewer 2 Report
Thank you for your time and effort to revise this manuscript.
Reviewer 3 Report
I have read the revised version of the manuscript and I think that many comments have been addressed. However, there are still things to be clarified.
The title - full stop is not necessary.
Abstract
- the multidisciplinary sample is mentioned several times. I suggest deleting "The results were drawn from a multi-disciplinary
sample, and represented a broad range of clinical settings".
Keywords: Peron-centred Practice; structural equation model, - typos and missing words
Introduction
The introduction seems to me still confusing in terms of the aim of the study and potential contributions to the literature. "This paper presents the outcomes from a programme of work that has focused on the development
of a conceptual framework for implementing person-centred practice, the Person-centred Practice Framework (PCPF) [10,-12], and development of a tool aligned to the framework that can offer a valid and reliable standardised measure, the Person-centred Practice" I am not sure that the study is about the development of a conceptual framework or about testing it.
The potential contributions to the literature are written in very general terms - the last sentence of the introduction.
Also, what is the difference between your study and previous study regarding this framework?
The name of the tables should be placed above the table.
Some typos I Table 3 - for example: ..321
The title Theoretical framework below Table 3 is not appropriate.
Relationship between Second Order Latent Variables - this section brings data, no sentences except for the last line. A more specific explanation is needed to help better understand the results.
The name of figure 2 should be placed below the figure. The typos should be corrected - for example <theoritical)
Discussion
"The findings from the study provide new data relating to the operalisation of person-centredness in healthcare". - this contribution is confusing.